# EduTubers's Pedagogical Best Practices and Their Theoretical Foundation

**Cynthia Pasquel-López * and Gabriel Valerio-Ureña**

Department of Humanities and Education, Tecnológico de Monterrey, Avda. Garza Sada 2501 sur, Col. Tecnológico, Monterrey CP64849, NL, Mexico
* Correspondence: a00609710@tec.mx

**Abstract:** (1) Background: The COVID-19 pandemic forced educational institutions to radically change their teaching and learning methods. Many institutions found it a challenge when they opted to deliver classes online. Given the success of several EduTubers, this study aimed to identify their best practices. (2) Method: The research was qualitative in nature and descriptive in scope and analyzed three angles, namely the perspective of EduTubers, their videos, and the assessment of the audience. Moreover, the results demonstrated that the level of awareness of these practices varies among actors. (3) Results: The study identified 12 practices divided into four categories, namely (a) resource management, (b) communication strategies, (c) content management, and (d) pedagogical strategies. (4) Conclusions: Followers were aware of the most evident practices, such as tone of voice and length of explanation, whereas EduTubers clearly convey practices such as anchoring in prior knowledge and use of associations. However, when analyzing the videos of EduTubers, the study found that they used other practices with low levels of awareness, such as mental representations, balanced use of different resources, or management of topics in a logical and hierarchical order.

**Keywords:** audiovisual materials; teaching practice; learning methods; educational innovation; YouTube





## 1. Introduction

The pandemic period has required educational institutions to radically change their teaching and learning methods by utilizing information technologies to overcome the barriers imposed by social distancing. In this context, a new lineage of producers of educational content is repositioning itself in the digital sphere: EduTubers. These are influencers of educational content, who students of different levels frequently and voluntary watch to learn different topics. YouTube classifies educational video channels as follows: (A) those focused on providing answers to common questions or general culture; (B) those intended to help enhance learning skills e.g., do it yourself; and (C) those created to assist with school assignments, which generally help in gaining school knowledge and skills.

YouTube records millions of connected users each month who consume more than one billion hours of video [1]. Research indicates that the main segment of its users is individuals aged between 15 and 35 years [2]. Educational channels contribute to this proportion, because, at this age range, many subscribers are in their stages of higher education. In addition, according to Newberry and Adame [3], 70% of millennial YouTube users as of 2018 used this application to learn new activities.

Alternatively, and paradoxically, capturing the attention and maintaining the motivation of students are frequently difficult for teachers under formal education mediated by technology. Moreover, new generations are increasingly demanding in terms of learning resources, giving increased importance to innovation and creativity, and integrating audio, movement, and visualization elements [4]. In this sense, avoiding a comparison of the success between various EduTubers and the difficulty experienced by many teachers in formal education is impossible.

Given the importance of this new platform of education, even for educational institutions, this present work focuses on investigating the practices of EduTubers that facilitate learning among their followers. The study assumes that, to the extent that it can identify practices that enable the capture of attention and that maintain motivation, it could provide suggestions for formal education in virtual initiatives (streaming) and in those initiatives that only use videos as educational resources.

Although studies on EduTubers have increased in recent years, research still tends to be concentrated on broad issues such as the profiles of EduTubers [5], the educational use of their channels [6–8], and communication-related themes such as EduTubers' communicative abilities and talents [9,10]. Because they concentrate on the best practices of EduTubers, the following studies are pertinent to the contemporary context: Pattier [11], who aimed to determine the influence of educational channels on society and to identify the key elements that contributed to the creation and dissemination of science videos, and López et al. [12], who intended to define the effective pedagogical practices used by EduTubers.

Many of the success factors observed and analyzed by Pattier [11,13] among EduTubers were related to the contents produced, such as short videos, explanatory videos with examples, experiences, or songs, use of international language, presentation of the objectives of the video, use of medium shots and normal angles, appearance of a person in the video, and the use of normalized and youthful language. Pattier [13] found other factors for the success of the EduTubers related to the use of social media platforms: YouTube channel statistics, the use of the YouTube platform and the use of social networks. Conversely, López et al. [12] synthesized the best pedagogical practices of two EduTubers as (1) using a simple language, (2) giving a paused explanation, (3) using examples and metaphors, (4) using a pleasant and respectful tone of voice, (5) good handwriting, (6) reflecting passion and positive energy to generate confidence, (7) communicating with a positive emotional approach, (8) appearing in the video, (9) promoting interaction and discussion with the audience, and (10) producing videos with original and attractive elements.

Finally, Brame [14] indicated that considering three elements, namely cognitive load, promotion of student engagement, and active learning, is vital to the effectivity of videos as teaching tools. Cognitive load refers to signaling with text or symbols to highlight important information, segmenting information to control the flow of new information, screening out extraneous information that does not contribute to the objective, and matching the modality using audio/verbal and visual/pictorial channels according to information presented. In terms of promoting student engagement, Brame suggests keeping the videos short, maintaining a less formal conversational language, speaking relatively quickly and with enthusiasm, creating materials that are perceived as personalized for a class, and providing visual elements that match the discussion. For active learning, the author recommends using guiding questions and interactive elements, integrating questions into the video, and making the videos part of a major topic or assignment. Brame bases these three elements on the cognitive load theories of Sweller et al. [15] and multimedia learning theory of Mayer [16], which both focus on information acquisition and processing channels. Recently, Fyfield et al. [17] highlighted the importance of designing videos by intuition instead of using an instructional design. In addition, the authors propose supporting active learning, benefiting holistic learning needs, and using pedagogical approaches.

However, the studies reviewed did not address aspects such as content management (e.g., objectivity of content), on-screen resource management (e.g., combining text, images, and narrative), or pedagogical strategies (e.g., use of mental representations or associations). These studies largely examined statistical data (e.g., frequency of video uploads), video editing (e.g., who and how they appear in the scene) and the use of the YouTube platform (e.g., management of the video list and video comments). Additionally, none of this research considers the theoretical foundations that would justify the effectiveness of such approaches. Finally, no Latin American or classroom-style EduTubers were used in any of these investigations.

To further comprehend the effective practices used by EduTubers, it would be crucial to discover other theoretical foundations in addition to those already discussed (Mayer and Sweller). Learning theories offer a framework for educational decision-making [18]. Behaviorism, cognitivism, and constructivism are the three main approaches [18]. On the one hand, behaviorism asserts that stimuli and reactions are how learning takes place (i.e., Watson, Pavlov, Skinner, and Bandura). According to the cognitivism movement, knowledge acquisition processes such as the succession of information presentation, memory and attention have relevance because learning is a mental activity that happens with the acquisition of facts, principles, and concepts to later apply them [19]. Contrarily, constructivism proposes that learning comes from within the individual through learning events that trigger thought to create new knowledge [18].

Other emerging theories are situated in social processes involving groups, social dynamics, and culture; and are related to the changing status and nature of work. More recent proposals of technology-based theories have been made in response to the need to study the transformation of learning, taking into consideration the new tools used in the teaching–learning process [20]. In other words, they are learning theories that, in some cases, can be adaptable and creative in promoting learning in many situations and informal settings. Consider connectivism, which implies that knowledge is spread through relationships with people (social networks) and the information they convey. Connectivism conceptualizes knowledge through the dissemination of information considering information and communication technology. The contextual learning theory of Lave and Wenger [21], or Siemens [22], holds that learning is a process that is created and positioned in a particular situation.

The objective of the current study was to identify the best practices of Spanish-speaking EduTubers to examine the fundamentals of learning theories that could support them. This study sought to answer the research objective from three perspectives, namely EduTubers, analysis of videos, and the assessment of the audience.

## 2. Materials and Methods

This research adopted qualitative and descriptive analyses to identify the best practices of EduTubers in maintaining the attention and motivation of content consumers. It used a sample that met the inclusion criteria and the maximum variation to select four channels. As a criterion, this research selected recognized EduTubers as generators of educational content. Sampling was performed using two methods proposed by Creswell and Poth [23]. The first was sampling by a criterion, i.e., EduTubers should be recognized in Google searches or should appear at the top of popularity lists and evaluations of educational content. The second sampling was maximum variation. In other words, EduTubers were selected based on audience focus, topics covered, and types of content produced.

### 2.1. Participants

The study identified 20 EduTuber channels, who were invited to participate. Responses were obtained from four channels, namely Math2Me, CuriosaMente, HappyLearning, and Pasos por la Ingeniería. These channels constitute the four cases examined in this research. First, the study conducted interviews and artifact analysis (videos and comments from followers). Second, content analysis was conducted on interviews and comments from followers. Lastly, the study employed an observation grid for the videos.

Table 1 displays the characteristics of each channel. Math2Me is a class-format channel created by two Mexicans, who have been creating tutorials for contests, mathematics topics, and high school and college entrance examinations for more than 10 years. Its content is mainly focused on high school students. CuriosaMente is a popular Mexican science channel, whose main contents are videos on knowledge or general culture. Although this channel aims to provide contents for children, it can cater to a wide audience. HappyLearning is a Spanish channel with school contents intended for children. The channel is a combination of science-oriented topics based on a curriculum established in the education

sector in Spain. Finally, Pasos por la Ingeniería is a Mexican class-format channel operating online for more than four years and which contains videos about engineering, physics and mathematics.

**Table 1.** Characteristics of the channels.

| Channel | Type of Content | Audience | Total Videos | Total Subscribers | Year they Joined YouTube |
|---|---|---|---|---|---|
| Math2Me | Mathematics | High school | 2948 | 2.07 million | 2009 |
| CuriosaMente | General knowledge | General | 304 | 1.68 million | 2013 |
| HappyLearning | Scholars | Elementary school | 339 | 1.28 million | 2015 |
| Pasos por ingeniería | Engineering | University | 795 | 192 thousand | 2015 |

Note: The data for the total videos and subscribers were collected in October 2020.

*2.2. Instruments*

The authors conducted semi-structured interviews to analyze the perspectives of EduTubers with a question base to collect their descriptions and interpretations and obtain the visions of each case [24]. This type of interview was considered due to the digital interview modality to manage time effectively. The interview was divided into four sections: general characteristics of the videos, creation process, elements and learning principles for the videos and audience considerations. The study used Zoom, a video conferencing tool, to conduct interviews with an average duration of 80 min. Before the interviews and data collection, the researchers sent a letter of consent via Google Forms to the four participants. Two of the channels agreed to use their real names, whereas the study assigned pseudonyms for the two other cases. For analysis, the study reviewed ten videos from each channel, which were selected using a specific criterion: videos that registered the highest number of views in the channels as of 30 October 2020. For all videos, the study used a grid, based on the literature, to record the best practices used in their elaboration and as mentioned by the creators. The study further noted other practices observed during the analysis. Subscriber comments on the videos were also used to review audience ratings. However, one channel disabled their comment feature, because children's videos, by law, must have their comments inactive. For details, see the Supplementary.

*2.3. Process and Analysis*

After the collection, the study obtained three types of information, namely the transcripts of the four interviews, observation grids of 40 videos, and comments from 353 subscribers on the videos to be examined. To analyze the interviews and subscriber comments, the study employed the spiral analysis strategy proposed by Creswell and Poth [23]. In the same manner, the study analyzed the videos by contrast and comparison to discern standard practices. Furthermore, a triangulation of information from interviews, videos, and subscriber comments was carried out, and the results were contrasted for validity [23]. In this manner, obtaining the reliability of information and ensuring the consistency and thoroughness of the study became possible [25].

The procedure for analysis of information was as follows. First, the interviews were transcribed. Subsequently, the study conducted the first coding by identifying evidence that could answer the research question. After defining the codes, they were integrated into an Excel spreadsheet for easy data manipulation. In Excel, the codes were screened to determine the categories in two waves to ensure a unified category per interviewee. Excel was used for the analysis of videos and subscriber comments. After examining the interviews to identify practices, subscriber comments were analyzed to determine the characteristics they value in an ideal educational video. The comments were collected and integrated to determine the resulting codes. Finally, the videos were examined using the observation grid to identify the practices mentioned by EduTubers and subscriber com-

ments, including other identified practices. In addition, the study reviewed the frequency (occurrence analysis) of each identified best practice.

### 3. Results

This section presents the results obtained by analyzing data collected through interviews, videos, and subscriber comments.

Table 2 presents the four categories of the best practices of EduTubers, namely resource management, communication strategies, content management, and teaching strategies. Resource management considers two practices, namely using clear and eye-catching resources and combining and/or dividing between text, images, and narration. Out of the two, only the first was present in the three scenarios analyzed (videos, interviews with EduTubers, and subscriber comments). The category communication strategies contained three practices, namely managing speed, tone, and clarity of voice, using colloquial language, and using informal communication. Out of these three, only managing speed, tone, and clarity of voice appeared in the three scenarios. In the case of content management, three practices are considered, namely using accurate and current content, supporting content with a scientific foundation, and presenting content in an interesting and impressive manner. The last two practices were detected in the three analyses. Finally, four practices were grouped under pedagogical strategies, namely explaining in a clear, concrete, orderly, and detailed manner, gradually increasing the complexity of explanations, making reference to previous knowledge and contexts familiar to viewers, and using mental representations and associations. In this case, only explaining in a clear, concrete, orderly and detailed manner was observed in the three analyses. In the interviews with the EduTubers, in general, they did not make significant reference to the use of pedagogical strategies.

**Table 2.** EduTubers' best practices found from video analysis (40), interview with EduTubers (4), and subscribers' comments (353).

| Practices | Analysis of Videos | Interviews | Subscriber Comments |
|---|---|---|---|
| Resource management | | | |
| Using clear and eye-catching audiovisual resources (image, text, and audio). | ✓ | ✓ | ✓ |
| Combining and/or divide between text, images, and narratives. | ✓ | × | × |
| Communication strategies | | | |
| Managing speed, tone, and clarity of voice. | ✓ | ✓ | ✓ |
| Using colloquial language. | ✓ | × | × |
| Using informal (peer-to-peer) communication. | ✓ | ✓ | × |
| Content management | | | |
| Sustaining topics with scientific bases and foundations (objectivity and veracity). | ✓ | ✓ | ✓ |
| Using accurate and current content. | ✓ | ✓ | × |
| Presenting content in an interesting and impressive way. | ✓ | ✓ | ✓ |
| Pedagogical strategies | | | |
| Making a clear, concrete, orderly and detailed explanation. | ✓ | ✓ | ✓ |
| Gradually increasing the complexity of the explanation. | ✓ | × | × |
| Referring to prior knowledge and contexts familiar to viewers. | ✓ | ✓ | × |
| Using mental representations and associations. | ✓ | × | × |

### *3.1. Resource Management*

### 3.1.1. Using Clear and Eye-Catching Audiovisual Resources (i.e., Image, Text, and Audio)

This practice refers to using resources that seek to generate amazement with a special attention to the necessary elements that capture the attention of an audience (i.e., sound and animation).

In this sense, José of the HappyLearning channel highlights several critical elements in their videos.

> On the other hand, if you give them those intonations, that the child himself says "wow", play a lot with the emphasis, right? and with all that, the music is fundamental. The image and the drawings and the animations are fundamental, I mean, in the end, it is a whole. Also, depending on the subject you are going to explain, you play with color, resources, or others. Mathematics, for example, if we are talking about division, hey, we take a pizza so that they understand that division is the most generous mathematical operation in the world, because it means dividing into equal parts and then we explain the arithmetic operation.

Likewise, in the interview with Marisol of Pasos por Ingeniería, she compared other channels where visual production is a priority. Having only a whiteboard as her primary visual tool, she stated that using colors that clearly convey to the audience the procedure performed and highlighting essential elements are essential. This strategy is one of the hallmarks of her channel.

> Another thing that people tell me in the comments . . . is that I should use colors. I think I was criticized for this in college, that "you are not in elementary school anymore" . . . And I think that with colors, I think it helps a lot with the kids' learning . . . I discovered that they learn more and that they copy it with the same colors . . .

On the other hand, the subscribers of the channels appreciate this practice. For example, a subscriber of CuriosaMente commented: "The illustration of CuriosaMente is definitely the best of all the knowledge channels." One subscriber of Math2Me, despite the use of only a whiteboard to illustrate a procedure, described a video as, "entertaining and attention-grabbing." Other more specific comments were made for Pasos por la Ingeniería, such as "very illustrative", "simple and readable", and "the colors help a lot".

### 3.1.2. Combining and Dividing between Text, Images, and Narratives

This practice pertains to the simultaneous balance and combination between different elements, such as text, images, and narratives.

The study found an example of this practice in one of the videos from CuriosaMente. The video "¿De dónde viene tu apellido?" ("Where does your last name come from?") illustrates a situation in which a straightforward graphic and text complemented the narrative without drawing the audience's focus on themselves. Therefore, the channel used representative images that do not require analysis as one listens to the narration. In addition, the channel used simple and clear text, avoiding dense text. In this manner, attention is appropriate to the content of the video.

### *3.2. Communication Strategies*

### 3.2.1. Managing the Speed, Clarity and Tone of Voice when Speaking

This practice denotes the importance of communication skills, especially the speed of speech, tone of voice, and clarity of pronunciation.

> For example, Andres and Claudia of Math2Me share several changes suggested by subscribers. Andres highlights the following: "a change that I think has come out of the last jobs and experience, mainly, obviously the tone of voice, right? I think we spoke faster before." Claudia adds: "Well no, if you look at the first ones, they were very low and it was as if he was embarrassed, it was like . . . "I'm

going to explain . . . over here" . . . it's something that, when you grasp it with practice, obviously right? the confidence of what you are doing".

Regarding subscriber comments, the study found that many of them praised Marisol's voice by describing it as beautiful, sweet, and melodious. For Math2Me, subscribers compared the teacher's voice with that of a very well-known and followed YouTuber in a positive manner.

### 3.2.2. Using Colloquial Language

Using colloquial language refers to using informal and straightforward language that is natural and spontaneous to create closeness and familiarity with the audience.

Another video from HappyLearning on the digestive system exemplified the use of common language to explain the function of certain organs and even to imagine their shapes. For example, the video states, "the esophagus is a tube that leads food to the stomach . . . Once they are mixed, they travel to the small intestine, the small intestine is a tube, as its name suggests, thin". In this practice, although the video mentioned technical terms, which may be less understood by children, a simple description helped to enhance understanding.

### 3.2.3. Using Informal (Peer-to-Peer) Communication

This method refers to the effort exerted to make students feel comfortable in watching videos and acquiring knowledge. In this manner, learning barriers between teachers and students are reduced by avoiding formalisms and encouraging horizontality.

For example, one of the creators of Math2Me shares one of the main objectives of the channel as well as the main element of their content:

In fact, the important part of what we were trying to do since the beginning, was to take away the seriousness of mathematics and to make it less formal. He almost wanted to wear a suit, well, to explain mathematics classes on video. Now we do not, I mean, we have to take that away because you are no longer in the classroom, you are no longer in the classroom, and the students then get lost among the technicalities . . . then it was also to change the way of communicating how you were going to give the class, to take that rigidity away from mathematics, from science, from education . . . but, not to treat it like math gods. However, to lose the fear, it was many, many years, it was the phrase we had in Math2Me, "lose the fear of mathematics", and that is what we try to do.

### *3.3. Content Management*

### 3.3.1. Sustaining Topics with Scientific Bases and Foundations (Objectivity and Veracity)

This strategy refers to the importance of presenting content based on scientific data or formal educational curricula.

Marisol of Pasos por Ingeniería indicated her primary sources for creating the channel's content and demonstrated that, in addition to books she selects to compare content and methods, she ensures that topics belong to a formal curriculum of an educational institution.

I choose a topic, that of the syllabus . . . I look at the curriculum, I see what the topics are to cover, then I select the bibliographies . . . I usually grab at least three books and more, with three, I consider that they are enough for mathematics because it is the same thing explained in different ways . . . I grab several books because we would think that one book already teaches well, well I have realized that no, then I take three. Because you also have to question what you know.

In the case of the importance given by consumers of these contents, the following comments made in CuriosaMente provide an example: "well-founded scientific basis", and "there are more videos like these, which are objective".

### 3.3.2. Using Accurate and Current Content

This practice denotes the importance of creating current and concise content for consumers by presenting relevant and current topics without sacrificing understanding.

For example, Marisol of Pasos por Ingeniería highlighted the difference between time allocated for a classroom session versus an explanation in a video on the Internet:

> First, you must have teaching skills, then to process it digitally, because if you copy the whole process that you take in the classroom digitally, it is impossible, it does not work like that, because you need something very concrete. The idea has to be very precise, and, in the classroom, we can digress, explain it more, even with your hands, things like that so you can expand for an hour or as long as your class lasts, but not on the Internet, it is concrete and what you choose to go to.

The videos are short; the average length of the videos is less than 10 min. In one channel, the average duration of the videos is 4 min. Therefore, the information presented has to be concise and accurate. Concerning the content and topics covered, the videos are relevant according to the audience of each channel. For example, in classroom-type channels, knowledge is based on a curriculum defined per academic level. Conversely, several science channels address current topics, such as COVID-19 or the war in Syria.

### 3.3.3. Presenting Content in an Interesting and Impressive Manner

This aspect refers to the importance of generating astonishment and opening abysses, motivating the audience with curious questions, reflecting a positive attitude, spreading joy, and bringing smiles to children and adults in such a manner that it generates learning through entertainment and emotions.

Tonatiuh of CuriosaMente narrated an experience during childhood, where he learned as he had fun or enjoyed a cartoon. He mentioned an interview with the founder of Disney, which led him to avoid minimizing the capabilities of the audience.

> Once I heard Disney, he was answering the question if he made movies for children; he said no, he made movies for that little piece we all had, where we have a space to wonder . . . I also like to think a little bit in the same way that we talk to that little piece of us with those curiosities, that maybe they were never solved, which is why he can reach children and grown-ups. Because that question of why the sky is blue, maybe we asked it when we were children, and they told us: "because it is so" and we never answered it and maybe when we grow up, we say again: "hey, really, why is the sky blue?".

Another example of this same category is the statement of José about the elements that videos should possess to enable knowledge to reach the child.

> The best way to educate children is with a smile, is to make them smile, because if you smile at the child, you already have him, he already has his heart, his head open, if you go with absolute seriousness, the child is going to stay like that (grimaces), you know, that is why it is fundamental. The smile is fundamental . . . the smile is the key . . . The way, the way is to make them smile or transmit a smile . . . it is the basic thing to start and to transmit, that is to say, the knowledge is there, but the vehicle with which you are going to take that knowledge to the essence of the child . . . to the child's interior, always must be with positivity, joy, smiles, and good vibes.

The study noted several subscriber comments related to this practice. One of the comments for Math2Me indicated that "it is very well understood and entertaining", whereas those for CuriosaMente described the content as follows: "I did not know some details of what was mentioned", "very interesting facts", and "it is impressive how they summarize it".

*3.4. Pedagogical Strategies*

3.4.1. Making a Clear, Concrete, Orderly and Detailed Explanation

This characteristic refers to the importance of making step-by-step videos that imply clarity, order, and detail, such that the audience can follow and understand the discussion.

In an interview with Claudia of Math2Me, she described the practices they have built for making videos, which have remained effective for them over the years.

> Explaining step-by-step, it is about not skipping obvious things because then that is where the students, even in the class, have the problem . . . that if you pass the X over here, then look where it is and where that element that you moved went, try obviously that . . . if it's a topic that you can explain in 5 min or less, you make it as short and as concise as possible so that the student then goes and learns what they have to learn.

The relevance of this practice is very notable in the subscriber comments for Curiosa-Mente, e.g., "everything is so synthesized and manageable", for Pasos por Ingeniería, e.g., "well explained", "clear, direct, and easy explanation", and "super well structured", and for Math2Me, e.g., "easy explanation" and "step-by-step explanation", where several subscribers in another video had requested a detailed explanation of a procedure.

3.4.2. Gradually Increasing the Complexity of the Explanation

This practice refers to the sequence of the contents presented in the video, such that they address the most complex topics at the end of the video. The idea is to gradually increase the complexity of the topics. In the same manner, information is presented in parts to avoid saturating the audience.

The study found an example of this practice in the video from Math2Me on factoring methods (Video: Método de factorización). The video individually and rapidly described seven factorization methods along with their explanations by flashing information on and off the screen. Additionally, they seemed to address a particular type of complexity in their presentation.

3.4.3. Referring to Prior Knowledge and Contexts Familiar to Viewers

This concept refers to the importance of including previous knowledge in explaining the current content, such that the audience can remember the knowledge and, thus, link it to newly acquired knowledge.

Tonatiuh, one of the partners of CuriosaMente, described a few of the most successful videos in the channel and their shared characteristics, such as a common theme and the approached used, in which the audience can also feel included in the interactive explanation.

> It seems to me that people have reacted more strongly to videos that involve them . . . as in their identity. For example, the one about who was the first human being and videos that have to do with, for example, if I am going to make a video about the evolution theory . . . I could say: there are traces of evolution in your body, right? and then I involve you personally, right, and let us see what I have in my body, what is a vestige of evolution . . . for example, one of the most successful ones was about names or what your name means and what your last name means which are relatively simple videos to make, but that involve people a lot . . .

Another example is the interview with Marisol of Pasos por Ingeniería, where she related her experience of taking university classes and one of the practices that she considers key to the explanation of her videos.

> I explain even laws of exponents of fractions and all that because many times I was told you go to high school and they say to you: you should have already learned this in elementary school when you go to high school; you should have already learned this, then when you are at university, this background is from high school . . . So, what I do, for example, if I am in an equation that has an

exponent and I tell them, this is "y" equal to the three. What does that mean? You must multiply three times the exponent, and many already know this because they have the background, but those who did not know it say, ay, look, this was never explained to me ... So that is very important ...

### 3.4.4. Using Mental Representations and Associations

This practice refers to the use of figures or animations that represent the information being shared and that facilitate the audience in making specific associations based on concepts or analogies that may be familiar to them.

To exemplify this practice, the video "El Sistema respiratorio" (Respiratory System) by HappyLearning illustrates a character taking a breath and appreciating the organs of the respiratory system at the same time. In this manner, the audience can imagine how lungs work when breathing by representing them in a cartoon.

In the same video, the illustration likens the mouth and nostrils to doors and windows with the following narration: "The airways are tubes through which air passes until it reaches the lungs. They start in the mouth and the nostrils, which are like doors or windows, where air enters our body when we breathe in".

## 4. Discussion

In summary, the study identified four categories with 12 practices observed in the analyzed YouTube educational channels which could positively impact the learning of viewers. On the foundations of traditional theories and the findings of recent studies, we can justify the effectiveness of these practices (conscious or unconscious) in the learning process.

EduTubers aim to prevent their followers from losing continuity of the explanation or diverting their attention to illustrations that do not contribute to the explanation. Sweller's cognitive load theory [15] could be used to support the practices related to the category of resource management. This theory implies that in order to comprehend a task and retrieve or retain information, the working memory requires a certain amount of mental effort. According to this theory, one should avoid overwhelming the audience with details or elements that do not support the explanation as it is being given. This study also discovered that EduTubers employ simple graphics and pictures together with adequate explanations. Additionally, they employ animations as appropriate to portray dynamic occurrences to better convey a new idea. Mayer's cognitive theory of multimedia learning [16] postulated learning as an active process through the auditory and visual channels, which have limited capacities to process information. This study found four of Mayer's proposed principles, i.e., spatial consistency, redundancy, modality, and multimedia, in the practices of EduTubers. These principles lead to, for instance, continuous text and images, a narrative to accompany the images, and the tuning of relevant representative images. Similar to these findings, previous authors highlighted the practices of using re-sources to make content more attractive [12,26], ensuring that the information used contributes to the stated objective, and combining visual elements with the explanation [14]. This research found that EduTubers intended to generate reactions and surprise viewers using attractive audiovisual resources. This strategy helps them spread positive emotions, capture the attention, and maintain the motivation of their followers. EduTubers give importance to the emotions generated in their channel and consider subscriber comments to modify their practices and contents. Guo and Fussell [27] conducted a study on emotional contagion in live streaming videos and presented a similar finding, i.e., the feelings inferred from the messages of the audience correlated with the narrative of the videos. Furthermore, Rosenbusch et al. [28] identified two forms of emotional contagion in YouTube videos: the transfer of emotions from (a) the video uploader and (b) subscriber comments.

Regarding the practices under communication strategies, EduTubers reported that these practices help them build a familiar environment that helps them connect easily with the audience. The principles of voice and personalization of cognitive theory of multimedia learning [16] are related to these practices. The first suggests that a familiar

and friendly tone can encourage learning, whereas the second postulates that people learn better when engaged in a conversational language. Utilizing a pleasant and courteous tone of voice, using normalized, young, and less formal language, and delivering information at a steady speed are all important considerations, according to López and Pattier [11–13]. Other research has shown that EduTubers employ a particular tone and rate of speech to emphasize crucial themes, even while using emotive language, to make the participants feel engaged and happy [29]. Similar to this, Brame and López et al. [12,14] emphasized the importance of speaking reasonably quickly while being enthusiastic and expressing passion and positive energy to inspire confidence. This statement coincided with that of Muñoz et al. [30] about the employment of narrative approaches in science-focused videos to engage viewers. This is consistent with what Tüzemen suggests [31], in that an atmosphere of trust and security is built with attitudes that allow them to connect through personal or sensitive conversations and intimate experiences. This set of practices reflects efforts to favor the concept that the narrator is a peer, not an authority figure.

According to this research, EduTubers avoid vertical narratives by using familiar topics, rescuing social learning processes, and considering the usage of youthful language. By referencing their own experiences and utilizing everyday language, these elements let the audience feel included in the learning process [11,12,28]. The study also found that social constructivism's guiding principles currently suggest that learning is generated internally, which requires social learning experiences that challenge thinking to create new knowledge [18]. According to Vygotsky [32], social interactions are the basis for learning. In other words, the social context serves as the main learning facilitator. EduTubers are concerned with keeping their content brief, which is consistent with Pattier's findings [11] and Brame's advice [14] regarding the importance of using short videos. When it comes to the practice of maintaining content with scientific foundations, Zaragoza and Roca Marín [29] highlight that making a video involves intense previous work to master and explain the topic of interest. Practices related to content management can be supported by Sweller's cognitive load theory [15] and the segmentation principle of Mayer's cognitive multimedia learning theory [16]. According to the segmentation principle, learning happens when information is broken up into manageable chunks [16].

The third category of content management, which examines the basis and fundamentals of the topics covered in the videos and how the information is delivered, can be supported by the rhetoric of logos, pathos, and ethos from Aristotle [33]. This presents the EduTuber as an authority on the subject (ethos) since they use reason and logic to persuade the audience intellectually. Finally, they add pathos so that the audience is emotionally attracted to the content, through surprise and curiosity.

The EduTubers frequently use the audience's existing knowledge to help them teach new material. This finding is similar to the result in López et al. [12], who proposed that EduTubers use examples and metaphors that facilitate content comprehension. In this sense, Ausubel's theory of meaningful learning [34] could support the category of the pedagogical strategies, which claims that learning happens when existing knowledge is connected to new information. Additionally, the practice of explaining in a clear, concise, organized, and detailed manner could be related to the segmentation principle of Mayer [16].

In general, the findings show that the best practices of EduTubers could be inspired by traditional learning theories and other emerging theories, even if unintentionally. This finding is consistent with Fernández-Cárdenas's [19] assertion that traditional learning theories continue to exist alongside new ones in modern education.

## 5. Conclusions

Given the abovementioned results, one of the first highlights of the current study is the level of awareness that different actors hold about the practices of EduTubers. In addition, this study found that the most apparent practices are conveyed in the subscriber comments, such as tone of voice, writing, and length of the explanation. As content producers, EduTubers are more sensitive to practices that may go unnoticed by the audience, such as

anchoring in prior knowledge, using associations, and using comparisons for enhanced learning. However, analysis of the videos indicated that EduTubers use other practices, such as using mental representations, balancing the use of different resources, or managing topic in terms of logical and hierarchical order. This study inferred that even the EduTubers were unaware of their observance of such practices.

On the other hand, in the results the co-creation between EduTubers and subscribers was evident, i.e., of a set of practices that seem to work in a virtual learning environment with short videos, with almost no interaction between the lessons. However, viewers seemed confident in evaluating the EduTubers' practices and content with the dilution of formality and authority figures that conventional schooling does not always make possible. At the same time, the EduTubers were willing to receive feedback and modify their practices that were inefficient to the teaching–learning process. In this manner, they adapted their content to be in line with the needs and demands of viewers. In addition, EduTubers seemingly unconsciously use certain learning strategies and modify their practices through trial and error.

As EduTubers have learned and improved their teaching practices during the COVID-19 pandemic and transformed the context of educational institutions, the extent to which several of their practices could be adapted to formal education warrants further investigation to generate added enthusiasm for knowledge, and to capture the attention and maintain the motivation of students. In this manner and despite a constant comparison with classroom teachers, as observed in subscriber comments, EduTubers do not consider their activities better than those of teachers due to the extremely different settings. For example, when looking for videos online to complement their education, students are motivated significantly differently than when they are in a classroom. Another example is the tangible connection established during learning activities, such that the level of interaction between teaches and students is extremely high in contrast to the work of EduTubers. Teachers could employ some of the practices used by EduTubers in the classroom, such as resource management, which shows how to combine the use of text, graphics, and audio for maximum effect. It would be challenging for a teacher to apply some practices, where there is a great need for video editing; nonetheless, it is not completely ruled out that the institution might assist in the production of such videos. Finally, it is important to consider if the systematic integration of theory-based practices in the content of EduTubers could increase the effectiveness of student learning while also maintaining their motivation and attention.

## 6. Limitations and Recommendations

Some implications of the present study are the differences in content between the channels studied, so the practices could be different from one type of content to another. It is also important to highlight the need to evaluate the content of educational videos on social platforms, some content may lack scientific rigor, as Mayo and collaborators [35] indicate that most online content of the medical field is neither useful nor safe; also Esparza and Sánchez [36] found that students base their evaluation on the number of likes and comments the video gets or by recommendations of people close to them rather on features related to the content. Furthermore, it is relevant to highlight that the use of this type of educational video and the number of educative channels on YouTube grew in the face of the pandemic; thus, it would be interesting to analyze those videos of teachers who have become EduTubers after the pandemic. It would also be fascinating to understand the transformation of teaching practices from the EduTuber's perspective and how these practices respond to the audience and/or align content with the economic drivers of YouTube [37]. Finally, the results of this study invite future research to identify the more valuable practices for certain type of content or channels. Finally, considering the unique circumstances of formal education, it could also be interesting to learn if the practices found would be as effective as they are in informal educational settings.

**Supplementary Materials:** The following are available online at https://www.mdpi.com/article/10.3390/informatics9040084/s1, S1: Interview, Table S1: Observation guide.

**Author Contributions:** Conceptualization, C.P.-L. and G.V.-U.; methodology, C.P.-L. and G.V.-U.; validation, C.P.-L. and G.V.-U.; formal analysis, C.P.-L. and G.V.-U.; investigation, C.P.-L. and G.V.-U.; data curation, C.P.-L. and G.V.-U.; writing—original draft preparation, C.P.-L. and G.V.-U.; writing—review and editing, C.P.-L. and G.V.-U.; supervision, G.V.-U.. All authors have read and agreed to the published version of the manuscript.

**Funding:** This research received no external funding.

**Institutional Review Board Statement:** Not applicable.

**Informed Consent Statement:** Not applicable.

**Data Availability Statement:** Not applicable.

**Acknowledgments:** We acknowledge the technical support of Writing Lab, Institute for the Future of Education, Tecnologico de Monterrey, Mexico, in the production of this work.

**Conflicts of Interest:** The authors declare no conflict of interest.

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
