# Peer review of "EduTubers’s Pedagogical Best Practices and Their Theoretical Foundation"

_informatics, doi:10.3390/informatics9040084_

Round 1

Reviewer 1 Report

Comments to the authors:

After the review of the article, it was found that the materials and method needs to be supported with clear and detail explanation about the participants and the process followed to reach the final findings.

With regards to the results and discussion parts, they are not in-depth. More clear details of the results obtained are needed. In the discussion part, it is important to try to related the previous literature and what the current study has contributed to our knowledge.

Author Response

Dear Reviewer,

Thanks for your valuable comments and recommendations, we have considered all of them to improve our work. In the attachment, you will find the changes to each recommendation.

Reviewer 2 Report

The topic of the manuscript addresses the examination of best practices among EduTubers. Taking into consideration the recent increase in online teaching methodologies, as well as the appearance of EduTubers as online teaching promoters, the topic might be of great interest to all foreign language teachers. In particular, I really enjoyed reading the manuscript. Nevertheless, there are several points that I believe need some improvement.

Title and Introduction

The title of the study itself seems to be too generic as the research question of the paper focuses on those best practices that promote attention and motivation.

In addition, the gap to be filled is not clear enough. As stated in the study, there are several investigations focusing on the same topic (best practices). Explain what your study adds to the previous research.

You may also have a look at some more recent studies: Daniel Pattier (2021), “Science on Youtube: Successful Edutubers” or Biadeni/Castro (2021) “Edutubers: reconfiguraciones de formas de enseñar a partir del uso de las plataformas de redes sociales”.

The objective of the current study is determined as “to identify the best practices of EduTubers in maintaining the attention and motivation of the consumers of their content”. Nevertheless, attention and motivation aspects are not mentioned in the theoretical background nor in the analysis and discussion sections.

Materials and Method

I would suggest dividing this section into subsections such as participants, tools, procedures, etc.

The instruments used for the study require a more detailed description. For example, for video analysis, describe the parameters that were taken into consideration and explain what research they were based on. The same happens with interviews. Explain the structure of the interview, the questions, etc. (You might include the documents as attachments).

In addition, coding and analysis procedures require a more detailed description. Otherwise, it is not clear where the four best practices categories come from.

Results

When describing your results, it might be interesting to separate those results coming from interviews from those coming from video analysis and comments.

Discussion

The discussion section is mostly based on justifying the best practices and relating them to the learning theories. However, I have not found anything that relates your findings to motivation theories. At least, this should be an expected outcome of the study (if we follow the objective announced in the introduction section).

Conclusions

The limitations of the study are not mentioned at all.

The pedagogical implications are announced on p.2 but are not further provided.

Finally, please, take into consideration that the manuscript requires language proofreading.

E.g.

Combining and/or divide between text, images, and narratives.  (p. 4 - Table)

In the video ‘De dónde viene tu apellido?’ shows a moment where a simple illustration 201 and text supported the narrative without monopolizing the attention of the audience. (p.5 , lines 201-202)

The justification of the efficiency of these practices (conscious or unconscious) in the learning process could be bases on principles of classical theories and on the results of recent studies. (p.8, lines 345-346)

Author Response

(The authors gave the same response as above.)

Reviewer 3 Report

Originality: excellent, contribution to the field: good, clarity of presentation: good, depth of research: good

The submitted article presents Identifying the EduTubers’s best practices. Author(s) provides original results of their investigations and examination of material from their own collections. The applied methods and the interpretation and presentation of results correspond to international standards. Findings are useful for educational, pedagogical, social environments. Author(s) have studied and used an appropriate number of bibliography sources. The language is not always perfect, the syntax is in some parts a bit convoluted. I recommend to add research problem, short implications of findings and short recommendation after conclusions – in my opinion it is obligatory to raise the level of the article.

Author Response

(The authors gave the same response as above.)

Round 2

Reviewer 2 Report

I appreciate the authors have taken into consideration most of my recommendations and suggestions. Nevertheless, there are still some issues that require further improvement.

The objective of the research and the gap you are trying to fill should be better tuned. Indicate in a clear way in which way your research deepens the works of Pattier and Lopez et al. (mentioned in the intro).  What are the limitations of these studies? And in which way do you pretend to improve their studies?

As regards the objective of your study, it seems that the aim of the study is to examine the theories that lie behind the best practices. If it is so, clearly state it and explain why it is important to research the theory of edutubers. Probably, the theory of learning is a rather well-known part and could be more logical to research how this theory is put into practice by Edytubers. Probably, your contribution to the topic might be to establish these connections (theory of learning and practices). In this case, you have to introduce the learning theories in your intro.

Anyway, this point should be clarified.  

The same doubt rises in the Discussion section as it is not clear what is your primary aim: theory or practice. I would suggest you reorganise this section connecting practices to theories and not vice versa. For example, you might start by describing your results (your best practices) and connecting them to learning theories etc.

The pedagogical implications of your study are not included at all. The implications mentioned in section 5 are rather limitations. Too much emphasis on further research.

The title of the research is too generic, I  think it also should be better tuned.

Still, the paper requires careful proofreading as there are some grammar, stylistic and spelling mistakes.

Author Response

Dear Reviewer,

Thank you for your valuable recommendations; we have considered all of them and we have made changes to the document.

Revisor recommendations

Changes

The objective of the research and the gap you are trying to fill should be better tuned. Indicate in a clear way in which way your research deepens the works of Pattier and Lopez et al. (mentioned in the intro).  What are the limitations of these studies? And in which way do you pretend to improve their studies?

The difference between the studies found are detailed in lines: 95-103

As regards the objective of your study, it seems that the aim of the study is to examine the theories that lie behind the best practices. If it is so, clearly state it and explain why it is important to research the theory of edutubers. Probably, the theory of learning is a rather well-known part and could be more logical to research how this theory is put into practice by Edytubers. Probably, your contribution to the topic might be to establish these connections (theory of learning and practices). In this case, you have to introduce the learning theories in your intro.

Learning theories are presented in lines: 104-126

The same doubt rises in the Discussion section as it is not clear what is your primary aim: theory or practice. I would suggest you reorganise this section connecting practices to theories and not vice versa. For example, you might start by describing your results (your best practices) and connecting them to learning theories etc.

We changed the discussion in lines: 467-546

The pedagogical implications of your study are not included at all. The implications mentioned in section 5 are rather limitations. Too much emphasis on further research.

We added in the conclusion section some pedagogical implications in lines: 560, 574-583

The title of the research is too generic, I  think it also should be better tuned.

In order to maintain a concise title, we used the following: EduTubers’s Pedagogical Best Practices and their Theoretical foundation. 

A more specific title could be: “Identifying the theoretical foundation of Spanish-speaker EduTubers’s pedagogical Best Practices”; however, we believe would be so large.
